# Auxiliary objective improves generalization performance but reduces model specification for low-data neuroimaging-based brain age prediction

**Donghyun Kim**
TReNDS center
Georgia Institute of Technology
dkim907@gatech.edu

**Eloy Geenjaar**
TReNDS center
Georgia Institute of Technology
egeenjaar@gatech.edu

**Vince Calhoun**
TReNDS center
Georgia Institute of Technology
vcalhoun@gatech.edu

## Abstract

Data scarcity and underspecification are 2 common issues in machine learning for healthcare. Data scarcity impedes the performance and generalizability of neural networks. Underspecification (where the training process can produce many different models that achieve the same train/test performance but represent different functions) may lead to issues with generalization and poor model behavior in deployment settings. In this work, we add an auxiliary objective to a brain age prediction model that significantly improves model performance and generalization in low-data regimes. We evaluate the impact of the auxiliary objective on model specification and particularly quantify how random variations in the training process affect a model's representations and predictions. Our results show that while the auxiliary objective enhances generalization and performance, especially in data-limited settings, it also reduces model specification. These findings underscore the trade-off between improving generalization with added constraints such as auxiliary losses, and their reduction in model specification in low-data neuroimaging applications.

## 1 Introduction

Data scarcity is a common issue in machine learning for healthcare due to the high costs associated with data labeling. Models trained on these small datasets demonstrate mixed performance and often do not generalize well [1]. A related issue is underspecification, where the training process can produce many different models that achieve the same train/test performance but represent different functions and thus focus on different features [2]. This makes it hard to select a model purely based on a model's validation performance. Underspecification can also affect generalization because each function may have learned features that do or do not transfer well. Given that these models may essentially exhibit the same performance, it is impossible to choose any model over the other. This can cause issues when models need to generalize to real-life datasets that are dissimilar from the training set.

Addressing data scarcity and underspecification will lead to more effective healthcare machine learning solutions across a broad range of modalities and applications. With the abundance of covariates that can have negative effects on algorithmic fairness in healthcare applications of AI [3],

it is especially important to increase the specification of neural networks in healthcare settings to ensure that the impact of potentially harmful covariates are reduced across all trained models.

In this work, we add an extra objective to a brain age prediction model's loss function to improve model performance and generalizability. We also assess whether the auxiliary objective leads to an increase in model specification. We focus on brain age prediction from 3D structural magnetic resonance imaging (sMRI) because it is commonplace in the field of neuroimaging [4, 5, 6], and has a clear ground truth. Our work in measuring two aspects of underspecification is impactful in other MRI prediction settings, such as Alzheimer's progression prediction [7], a clinically important field of study where neural networks have become commonplace.

**Background** In auxiliary learning, a second objective (auxiliary task) is added to a neural network's loss to help guide it toward consistently learning relevant features. Specifically, previous work [8] has proposed a method to automatically reweigh auxiliary tasks by reducing the distance between gradients of the two loss terms. Follow-up work [9] applied the adaptive reweighing scheme to improve the performance of multiple myeloma classification in data scarcity settings.

Some studies have also proposed methods to mitigate underspecification. Distilling features learned from multiple models into a global model can help improve out-of-distribution performance [10]. Moreover, using minimal preprocessing, model regularization, and data augmentation can enhance generalizability and robustness in brain age prediction [11]. However, no works have looked at either improving brain age prediction with an auxiliary objective function or the evaluation of both generalization and underspecification performance. This is important because, in real-life settings, it is essential to be able to pick a specific model, even under training variations (e.g. random initializations).

## 2 Method

**Model** To study underspecification in neuroimaging, we use a simple yet powerful model called the simple fully convolutional network (SFCN) [4]. The original model consists of 7 modules, where the first 5 modules are made up of a 3 x 3 x 3 3D convolution, batchnorm, maxpool, and a ReLU activation. The 6th module consists of a 1 x 1 x 1 3D convolution, batchnorm, and a ReLU activation. The final module consists of an average pooling, dropout, 1 x 1 x1 3D convolution, and a softmax. We remove the maxpool layers from the 5th module to accommodate the size of our preprocessed data. The first 6 modules are feature extraction layers. The final module linearly maps the model's representations to a probability distribution over the ages. The SFCN model [4] minimizes a Kullback–Leibler divergence loss between the predicted probability and a Gaussian distribution centered at the true age with a standard deviation of 1.

**Measuring underspecification and generalization** First, to measure underspecification, we calculate the similarity between model representations and predictions across random initializations and training data variations. In this case, representations refer to activations from the last feature extraction layer, before a prediction is made with the last 1-by-1-by-1 convolutional layer. We calculate the linear de-biased pairwise centered kernel alignment (linear CKA) [12] between pairs of representations of the same subject from the same model trained with different initializations and different training distributions. We also calculate the average absolute difference of age predictions of the same subject made between pairs of the same model trained with different initializations and training distributions.

Models are more specified if there is less variation in the representation space (indicated by higher CKA) and when predictions are more similar (indicated by a lower average absolute difference). Second, to measure generalization performance, we test how model performance generalizes to a new data distribution with unseen age ranges.

**Auxiliary learning** An auxiliary task is a second objective that is added to a neural network's loss to help guide it toward consistently learning relevant features. In low-data regimes, a model may find it difficult to learn to find the same global minimum across different runs, resulting in underspecification. To help the model, we add an easier classification objective. The auxiliary loss we propose forces the model to predict one of five quantiles of a subject's age. Specifically, we add a linear map (a 1-by-1-by-1 3D convolution) that maps the model's representation to 5 logits. The

auxiliary loss is calculated as the negative log-likelihood of the logits under the target age's quantile. This loss is added to the Kullback–Leibler divergence loss from the SFCN model.

With the auxiliary loss term, the model is not only pushed to learn the exact age but also to predict a rough guess of a person's age. We hypothesize that this improves performance in low-data settings because the model may converge to the auxiliary loss better when few subjects are available.

For underspecification, the auxiliary loss can either increase model specification by smoothing the loss landscape or decrease model specification by increasing the number of local minima. Since the minimum for the original loss function is included in the minimum of the auxiliary loss, i.e. predicting someone's age is harder than predicting their general age range, the loss landscape may become smoother. However, adding multiple objectives also makes the training dynamics less predictable, and can add noise to the training process that may exacerbate underspecification. We explore the effect of the auxiliary task on model specification.

For the remaining sections, let an auxiliary model be a model that uses a sum of the Kullback–Leibler divergence loss and auxiliary loss as its loss term. Let a baseline model be a model that uses only the Kullback–Leibler divergence loss as its loss term.

# 3   Experiments

**Dataset and experimental settings**   Following the SFCN paper, we use the UKBiobank dataset, but with an SPM pre-processing pipeline that follows previous work [13]. Each sMRI volume after pre-processing is 121-by-145-by-121 voxels and is smoothed with a 6mm full width at half maximum (FWHM). To select the training and validation data, we use stratified sampling based on the 5 quantiles of the age in the full dataset (N=37852). Each training/validation set is split 80/20 respectively. Both the auxiliary and baseline models are run across 5 training/validation set sizes, $\{100, 250, 500, 1000, 2000\}$, 10 random seeds, $\{0, 1, 2, 3, 4, 5, 6, 7, 8, 9\}$, and 5 randomly sampled folds. This results in $10 * 5 * 5 = 250$ runs for both the baseline and auxiliary model (500 runs total). Each instantiation of a model is trained for 200 epochs. As a test set, we use 5200 unseen volumes (200 volumes per age) with a uniform age distribution. Although the original training set is distributed according to the proportion of the 5 quantiles of ages in the full dataset, our test set has a uniform distribution to thoroughly test model performance and model specification on each age. Additionally, we create an out-of-distribution dataset that consists of 3400 unseen volumes, 100 volumes per age, with unseen age ranges. Specifically, the training, validation, and test set consists of ages that range from 50 to 75, and the out-of-distribution set consists of ages that range from 47 to 80.

**Auxiliary learning improves model performance**   To understand how well the models perform on a normal generalization task, we evaluate the MAE (mean absolute error) on the test set. Especially when little data is available, as shown in Table 1, the auxiliary model significantly outperforms the baseline model. Specifically, for data regimes with a size less than 2000, the auxiliary model outperforms the baseline model. We see that the gap between the MAE of the auxiliary model and baseline model is greatest when 250 samples are available. Interestingly, the gap between the models with 100 samples is minimal. For this data regime, it's likely hard to converge for both the baseline and auxiliary models.

Table 1: Average MAE. Each significance result is based on a paired two-sided t-test. Values where the auxiliary model performs significantly better than the baseline model are made bold.

| Train/val size | Auxiliary model | Baseline model | p-value |
|---|---|---|---|
| 100 | **6.1136** | **6.1814** | **3.70E-2** |
| 250 | **5.0248** | **5.172** | **2.54-4** |
| 500 | **4.5513** | **4.6713** | **8.69E-4** |
| 1000 | **3.8176** | **3.9455** | **1.51E-14** |
| 2000 | 3.4521 | 3.4489 | 6.76E-1 |

**Auxiliary learning improves out-of-distribution predictions**   To test whether the representations the model has learned can generalize to unseen age ranges, we create an out-of-distribution dataset with unseen age ranges and volumes. Each volume in this out-of-distribution set is embedded into a

representation, and we train an ElasticNet model on the representations by splitting the representations into a stratified training and test set (80/20 ratio). The ElasticNet performs cross-validation across 5 folds on the training set. As shown in Table 2, for train/val sets with fewer than 1000 samples, the auxiliary model significantly outperforms the baseline model both on out-of-distribution ages specifically, and across all other unseen volumes as well.

Table 2: Mean R-squared score, predicting out of distribution ages. Each significance result is based on a paired two-sided t-test. Values where the auxiliary model performs significantly better than the baseline model are made bold.

| Data regime | Predicting all ages | | | Predicting OOD ages | | |
|---|---|---|---|---|---|---|
| | Aux | Baseline | p-value | Aux | Baseline | p-value |
| 100 | **0.512** | **0.497** | **0.0656** | **0.578** | **0.557** | **0.0684** |
| 250 | **0.629** | **0.61** | **1.29E-10** | **0.746** | **0.722** | **1.81E-09** |
| 500 | **0.669** | **0.648** | **3E-10** | **0.791** | **0.769** | **1E-12** |
| 1000 | **0.736** | **0.724** | **4E-7** | 0.841 | 0.837 | 0.175 |
| 2000 | 0.766 | 0.768 | 0.0468 | 0.858 | 0.864 | 9E-06 |

**Auxiliary learning reduces model specification across random initializations**    To calculate model specification across random model initializations, we embed all volumes from the test set to obtain the model's representations and age predictions. Then, we calculate all pairwise similarities (linear CKA) between representations of the same subject from the same model but trained with different random initializations. Additionally, we calculate the pairwise absolute difference between predictions of the same subject from the same model but trained with different random initializations. As shown in Table 3, for train/val sets of size greater than 250, the auxiliary model's representations and predictions are less similar across random seeds. This indicates that the model is less specified.

Table 3: Average similarity between the same models trained with different random initializations. Similarity is both calculated on the model's representations (CKA) and predicted age (average absolute difference). Values where the auxiliary model performs significantly worse than the baseline model are made bold.

| Data regime | representation CKA ↑ | | | predicted age (avg absolute diff) ↓ | | |
|---|---|---|---|---|---|---|
| | Aux | Baseline | p-val | Aux | Baseline | p-val |
| 100 | 0.9380 | 0.9374 | 0.95 | 0.4616 | 0.4278 | 0.397 |
| 250 | 0.9781 | 0.9684 | 1.25E-3 | 0.4725 | 0.4958 | 0.302 |
| 500 | **0.9764** | **0.9847** | **3.78E-19** | **0.4983** | **0.4624** | **8.36E-3** |
| 1000 | **0.9594** | **0.9739** | **9.91E-61** | **0.7588** | **0.5879** | **5.90E-61** |
| 2000 | **0.9489** | **0.9635** | **2.86E-58** | **0.9081** | **0.7643** | **4.04E-73** |

**Auxiliary learning reduces model specification across training folds**    To calculate model specification across training distributions, we embed all volumes from the test set to obtain the model's representations and age predictions. Just as in the previous experiment, we calculate all pairwise similarities (linear CKA) between representations of the same subject from the same model but trained with different folds. Next, we calculate the pairwise absolute difference between predictions of the same subject from the same model but trained with different folds. As shown in Table 4, for train/val sets of size greater than 100, the auxiliary model's representations and predictions are less similar across random seeds. This indicates that the model is less specified.

Table 4: Average similarity between the same models trained with different training distributions. Similarity is both calculated on the model's representations (CKA) and predicted age (average absolute difference). Values where the auxiliary model performs significantly worse than the baseline model are made bold.

| Data regime | representation CKA ↑ | | | predicted age (avg absolute diff) ↓ | | |
|---|---|---|---|---|---|---|
| | Aux | Baseline | p-val | Aux | Baseline | p-val |
| 100 | **0.8841** | **0.9136** | **0.045** | 0.7086 | 0.8917 | 1.13E-3 |
| 250 | **0.9302** | **0.9512** | **7.11E-6** | **0.7803** | **0.6659** | **3.58E-4** |
| 500 | **0.9115** | **0.9649** | **5.14E-37** | **1.0416** | **0.8570** | **8.53E-07** |
| 1000 | **0.8638** | **0.9174** | **1.88E-39** | **1.4249** | **1.1257** | **7.82E-23** |
| 2000 | **0.8469** | **0.8848** | **4.78E-52** | **1.5902** | **1.3555** | **5.42E-51** |

## 4 Conclusion

This work explores the utility of an auxiliary objective added to brain age prediction models. We find that while the auxiliary objective improves model performance and generalization to out-of-distribution data, the objective also amplifies underspecification. Specifically, models trained with an auxiliary loss exhibit more variability in both learned representations and predictions across different random initializations and training folds. Thus, our hypothesis underlining the added noise introduced in the training dynamics is more likely to be true. Future work should continue to explore methods that both increase model specification and model generalization. Our code is available at `https://github.com/donghyunkm/multitaskNeuro`.

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
