# OpenReview forum: "Auxiliary objective improves generalization performance but reduces model specification for low-data neuroimaging-based brain age prediction"
_NeurIPS.cc/2024/Workshop/UniReps — UniReps_

### Official Review · Reviewer_MNv1 · 2024-10-06
**Review of Brain Age Prediction via Auxiliary Objectives: Strong Generalization Gains in Low-Data Settings, but Trade-offs in Model Specification and Statistical Rigor**

**Rating:** 5
**Confidence:** 3

**Review:**

The paper addresses the challenge of data scarcity in healthcare applications, particularly focusing on brain age prediction from 3D structural magnetic resonance imaging (sMRI). It proposes adding an auxiliary objective to improve model generalization and performance in low-data settings, but also investigates the impact on model specification. Using a well-known fully convolutional network (SFCN), the paper demonstrates that while auxiliary objectives significantly enhance generalization, they simultaneously reduce model specification. The authors employ multiple experiments with stratified sampling on the UK Biobank dataset to evaluate this trade-off. The paper contributes to the ongoing conversation around balancing generalization and model specificity in healthcare AI, especially in data-limited environments.

Strengths

	•	Generalization in Low-Data Settings: The primary strength of this paper is its clear contribution to generalization in low-data regimes. The use of auxiliary objectives enhances the model’s ability to perform well, even when the training data is limited. This is an essential consideration in healthcare applications, where large datasets are often unavailable, and the ability to generalize to new, unseen data is critical.
	•	Thorough Experimental Design: The paper presents a well-executed experimental setup, utilizing the UK Biobank dataset with multiple training sizes, random seeds, and validation approaches. The evaluation of both within-distribution and out-of-distribution generalization provides a comprehensive understanding of the performance of both the auxiliary and baseline models.
	•	Impact on Neuroimaging Models: This work addresses a highly relevant and specific problem in healthcare AI, focusing on brain age prediction from 3D sMRI data, which is a clinically important task. Improving generalization for such tasks can have a direct impact on clinical settings, potentially leading to better diagnostic tools.

Weaknesses

	•	Model Specification Issues: One of the paper’s major weaknesses is the reduction in model specification when auxiliary objectives are introduced. While improving generalization, the auxiliary model produces greater variability across different random initializations and training folds. This variability could be problematic in clinical applications, where consistent predictions are vital.
	•	Lack of Statistical Rigor: The statistical testing methodology (paired t-test) raises potential concerns. The paper does not discuss whether the normality assumption was checked, which is crucial for the reliability of the p-values. If the performance metrics (such as MAE) are non-normally distributed, the significance results could be questionable. This could affect the validity of the claimed improvements.
	•	Limited Novelty: While the paper introduces an interesting approach to auxiliary learning, the core ideas are not entirely novel. Auxiliary tasks have been explored in other medical imaging tasks, such as domain adaptation and few-shot learning, as seen in other works like [Embracing the Disharmony in Medical Imaging: A Simple and Effective Framework for Domain Adaptation, Wang et al.]. The paper’s main contribution is the application of these ideas specifically to brain age prediction, but the methodological novelty is somewhat limited.

Questions

	1.	Can the authors provide further empirical evidence or theoretical justification for why auxiliary tasks lead to such variability in model specification? Would more complex regularization techniques help mitigate this?
	2.	Could the authors compare their method against other auxiliary learning frameworks in medical imaging, such as those used in cancer detection or Alzheimer’s disease prediction?
	3.	How generalizable is this approach to other neuroimaging-based tasks or different medical imaging modalities?
	4.	Did the authors check the normality assumption before using the paired t-test? If not, would they consider using a non-parametric test or performing a normality test to confirm the reliability of the statistical significance claims?

Limitations

	•	The paper addresses limitations clearly, particularly regarding model specification issues with auxiliary objectives. However, more potential solutions could be offered, such as ways to mitigate the negative effects of underspecification (e.g., through better regularization or model selection strategies).
	•	The potential societal impact of underspecification in healthcare applications, where consistency and reproducibility are critical, should be discussed more thoroughly. How might variability across model runs affect clinical decision-making?

---

### Official Review · Reviewer_nXmW · 2024-10-06
**Interesting paper but lacking some details**

**Rating:** 6
**Confidence:** 4

**Review:**

Summary: The paper considers the underspecification problem, a problem where multiple models can fit to the train/test data while using different features, and its application in neuroimaging. The paper considers the common approach of auxiliary learning, where a second objective is added to model training to incorporate specific features in model learning which has been used to improve generalization. The paper applies this technique to neuroimaging to characterize the relationship between underspecification and generalization with auxliary learning. The paper finds that the auxiliary learning task improves generalization for OOD ages in neuroimaging but reduces model specification.

Strengths:
1. The paper answers a pretty relevant question about the specific nature of an auxiliary task and shows a pretty surprising result that auxiliary learning makes a model less specified.
2. The paper's results establish a pretty important problem where the ability to generalize may not imply real-world applicability since it's potentially unclear what features a model is using.
3. The paper applies their technique for brain age prediction and finds improved generalization results for neuroimaging, which could be an important problem.

Weaknesses:
1. In general, I found the paper hard to read and the introduction hard to understand. The paper could do a better job introducing the problem it wants to solve. I found myself having the reread the introduction to get a sense of what the problem statement was.
2. The results of the paper are narrow and I wonder if this kind of result can be generalized beyond neuroimaging. This is a nitpick though.

Questions:
1. Did using a convnet influence your results? I wonder if comparisons across different kinds of models would maintain these results for your representation space?
2. Did you consider other kinds of metrics other than CKA for comparing representations?

---

### Official Review · Reviewer_yK36 · 2024-10-07

**Rating:** 5
**Confidence:** 3

**Review:**

The paper studies the trade-off between generalization and underspecification when using auxiliary objectives, particularly in the domain of medical image. The paper is significant as the data scarcity and model underspecification is prevalent, in neuroimaging and the paper aims to address important problems.

One suggestion for future improvement is the clear articulation of the main claim of the paper. In its current form, it is somewhat confusing whether the main goal of the paper is to improve generalization or analyze the underspecification.

---

### Decision · Program_Chairs · 2024-10-10

**Decision:**

Accept

**Comment:**

In light of the reviewers' feedback and relevancy of the submission, we are pleased to accept this paper for presentation at UniReps 2024. We kindly ask the authors to incorporate the reviewers' suggestions and feedback in the final camera-ready version of the manuscript.